# Familial Risks for Liver, Gallbladder and Bile Duct Cancers and for Their Risk Factors in Sweden, a Low-Incidence Country

**DOI:** 10.3390/cancers14081938

**Published:** 2022-04-12

**Authors:** Kari Hemminki, Kristina Sundquist, Jan Sundquist, Asta Försti, Vaclav Liska, Akseli Hemminki, Xinjun Li

**Affiliations:** 1Faculty of Medicine and Biomedical Center in Pilsen, Charles University in Prague, 30605 Pilsen, Czech Republic; 2Division of Cancer Epidemiology, German Cancer Research Center (DKFZ), Im Neuenheimer Feld 580, D-69120 Heidelberg, Germany; 3Center for Primary Health Care Research, Lund University, 205 02 Malmö, Sweden; kristina.sundquist@med.lu.se (K.S.); jan.sundquist@med.lu.se (J.S.); a.foersti@dkfz.de (A.F.); xinjun.li@med.lu.se (X.L.); 4Department of Family Medicine and Community Health, Icahn School of Medicine at Mount Sinai, New York, NY 10029, USA; 5Department of Population Health Science and Policy, Icahn School of Medicine at Mount Sinai, New York, NY 10029, USA; 6Center for Community-Based Healthcare Research and Education (CoHRE), Department of Functional Pathology, School of Medicine, Shimane University, Shimane 693-8501, Japan; 7Hopp Children’s Cancer Center (KiTZ), D-69120 Heidelberg, Germany; 8Division of Pediatric Neurooncology, German Cancer Research Center (DKFZ), German Cancer Consortium (DKTK), D-69120 Heidelberg, Germany; 9Department of Surgery, School of Medicine in Pilsen, University Hospital, Charles University, 30605 Pilsen, Czech Republic; vena.liska@skaut.cz; 10Biomedical Center, Faculty of Medicine and Biomedical Center in Pilsen, Charles University in Prague, 30605 Pilsen, Czech Republic; 11Cancer Gene Therapy Group, Translational Immunology Research Program, University of Helsinki, 00290 Helsinki, Finland; akseli.hemminki@helsinki.fi; 12Comprehensive Cancer Center, Helsinki University Hospital, 00290 Helsinki, Finland

**Keywords:** hepatocellular carcinoma, gallbladder cancer, risk factors, spouse correlation, discordant familial risks, alcohol

## Abstract

**Simple Summary:**

Familial risk of cancer implies that two or more family members are diagnosed with the same cancer. This may be due to the genes or environmental factors that family members share. Familial risk for liver and gallbladder cancer is about 2.7 which means that when one family member is diagnosed with these cancers other family members have 2.7 times higher risk of being diagnosed with the same cancers compared to families were no member is yet diagnosed with these cancers. Risk between spouses is entirely due to shared environmental factors and for liver cancer there is a small risk. The most important way to prevent these cancers is to avoid their risk factors, alcohol, smoking and overweight, and to seek medical care for diabetes and liver infections.

**Abstract:**

We used the Swedish Cancer Registry data to address familial risks for concordant (same) and discordant (different) hepatobiliary cancers, including their associations with any other cancers and with known risk factors. Risks were also assessed between spouses. The analysis covered Swedish families and their cancers between years 1958 and 2018. Adjusted familial risks were expressed as standardized incidence ratios (SIRs). Familial SIRs for concordant hepatocellular carcinoma (HCC) were 2.60, and for gallbladder cancer they were at the same level (2.76). Familial risk was also found for intrahepatic bile duct cancer and for female extrahepatic bile duct cancer. HCC was associated with lung and cervical cancers; extrahepatic bile duct and ampullary cancers were associated with colon and pancreatic cancers, suggesting Lynch syndrome. Among spouses, hepatobiliary cancer was associated with HCC, stomach, pancreatic, cervical and upper aerodigestive tract cancers. Among risk factors, family members diagnosed with alcohol-related disease showed association with HCC. The observed familial risks for hepatobiliary cancers were relatively high, and considering the poor prognosis of these cancers, prevention is of the utmost importance and should focus on moderation of alcohol consumption, vaccination/treatment of hepatitis viral infections and avoidance of overweight and other risk factors of type 2 diabetes.

## 1. Introduction

Primary hepatobiliary cancers include hepatocellular carcinoma (HCC), gallbladder cancer (GBC), cancers of the biliary tract (extrahepatic and intrahepatic bile ducts, also called cholangiocarcinoma) and of the ampulla of Vater (also called ampullary cancer) [1,2]. Risk factors for these cancers are many and the large international variation is in part explained by known risk factors [1,2]. Northern European risk factors of HCC include alcohol-related liver cirrhosis, chronic infection by hepatitis B (HBV) or C virus (HCV), obesity, low physical activity, type 2 diabetes, non-alcoholic metabolic liver disease, smoking, autoimmune disease and family history [1,2,3,4,5,6,7,8,9]. All hepatobiliary tract cancers were increased after exposure to Thorotrast, a widely used radiographic contrast agent between the 1930s and 1950s, including the radioactive compound thorium dioxide, producing excellent images but harmful alpha radiation as it decays [10]. According to a Swedish national HCC register, covering the years 2009–2016 and 3376 patients, the type of underlying liver disease was caused by HCV in 27%, alcohol in 23%, HBV in 5% and diabetes and non-alcoholic liver disease in 4% each of HCC patients [11]. For GBC, gallstones are an important risk factor, and others include bacterial infections, obesity, family history, biliary cysts and other structural abnormalities [1,4,12]. Biliary tract cancers share many risk factors with HCC but associations with HBV, HCV and alcohol are weaker [1,13,14].

Familial cancer can be caused by shared (germline) genetic or environmental risk factors. The latter may be estimated by shared cancer risk between spouses, which has shown no increase for liver cancer in a Swedish study [15]. However, familial risk may also be transmitted through an intermediary familial condition, which is a risk factor for cancer. For example, cholelithiasis predisposes to GBC but it has an independent familial component [16]. Other such conditions are diabetes and obesity. The large literature on familial HCC includes also studies on families of HBV-infected individuals, as reviewed in [17,18]. Fewer family studies have been published on GBC and biliary cancer but the relative risks, at about 3.0, have not been very different from results on HCC [4,19,20,21]. Probably because of the multiplicity of environmental risk factors, germline genetics of hepatobiliary cancers has been most successful in identifying low-risk variants [22,23]. A limited number of high-risk genes have been identified. GBC and biliary tract cancers manifest in hereditary nonpolyposis colorectal cancer syndrome (HNPCC, Lynch syndrome), particularly in mutation carriers of the mismatch repair gene MLH1 [24,25]. Low frequencies of mismatch repair gene mutations have been reported also in HCC [26]. BRCA1/2 mutations have been described in bile duct cancers [27].

We use data from the Swedish Cancer Registry to analyze familial risk in defined hepatobiliary cancers when family members were diagnosed with the same cancers or any other specified cancers. Additionally, risks for hepatobiliary cancers were assessed when family members were diagnosed with any of 12 risk factors for hepatobiliary cancers. Cohort design and completeness of the data, including familial relationships, are unique advantages of the applied setting. With this sample size, we can address familial risks even in rare GBC and bile ductal cancers.

## 2. Materials and Methods

The most recent update (latest data from 2018) of the Swedish Cancer Registry was used in the study. Family relationships were obtained from the Multigeneration Register, containing the Swedish population in families [28]. ‘The offspring generation’ was born after 1931 and by 2018, the oldest offspring reached the age of 86 years; their siblings could be defined only in the offspring generation. Cancers were identified from the Swedish Cancer Registry which was started in 1958 using codes of the International Classification of Diseases (ICD) versions 7 code 155 for all primary hepatobiliary cancer (155.0 HCC, 155.1 GBC, 155.2 bile ducts, 155.3 ampulla of Vater), but excluding subcodes 155.8 (cancer in multiple bile ducts) and 155.9 (cancer in unspecified bile ducts). Since the ICD-7 code 155.0 for HCC also included intrahepatic bile duct cancer, we additionally used ICD-10 from year 1997 onwards where these cancers were separated. Information from the registers was linked at the individual level via the national 10-digit civic registration number. In the linked dataset, civic registration numbers were replaced with serial numbers to ensure anonymity.

Familial risk was considered for offspring whose first-degree relatives (parents or siblings) were diagnosed with the same (concordant) or different (discordant) cancer; the relatives were thus probands. Standardized incidence ratios (SIRs) were calculated as the ratio of observed to expected number of cases. The expected numbers were calculated from the present dataset for all individuals without cancer among their family members (i.e., most of the Swedish population), and the rates were standardized by 5-year age, gender, period (5-years group), highest educational level (as proxy for socioeconomic status) and geographical region. The 95% confidence interval (95% CI) of the SIR was calculated assuming a Poisson distribution. Observed cases (O) indicate the persons whom the SIR was calculated. Spouses were identified through the first common child of the couples.

Patients diagnosed with any of ICD codes for 12 risk factors of hepatobiliary cancer (ICD codes for the risk factors are shown in **Appendix A**) were identified from the Swedish Inpatient Register (started in 1964 reaching nationwide coverage in 1987) and the Outpatient Register (nationwide since 2001) [29]. Each patient was entered only once for their first diagnosed morbidity. Analysis was conducted as described above.

Age-specific incidence rates were calculated between familial and nonfamilial cases of HCC and GBC. The rate ratios were plotted to show the age dependence of familial risk.

The analysis scheme is shown in **Figure 1**, highlighting the study subjects, the types of analysis (familial and spousal) and the diseases considered.

The study was approved by the Regional Ethical Review Board in Lund 6 February 2013 (Reference 2012/795 and subsequent amendments). Guidelines of the Helsinki Declaration were followed. The study was conducted in accordance with the approved guidelines with explicit statement that no informed consent was required. The study is a national-register-based study on anonymous personal data.

## 3. Results

The characteristics of the study population are described in **Appendix A**. The total offspring generation index population at age 20–86 years amounted to 9.34 million individuals recorded from 1932 to December 31, 2018. Hepatobiliary cancer was diagnosed in 8558 patients without family history (mean diagnostic age 61.7 years) and in 304 familial cancer patients (mean diagnostic age 63.9 years). Familial HCC was identified in 189 patients, GBC was identified in 76 patients, extrahepatic bile duct cancer was identified in 29 patients and ampullary cancer was identified in 10 patients. From these figures, one can calculate that the familial proportion (i.e., familial cases of all cases in offspring) of these cancers was 3.4% for all hepatobiliary cancer, 3.6% for HCC, 4.0% for GBC, 2.5% for extrahepatic bile duct cancer and 2.0% for ampullary cancer.

Familial risks for hepatobiliary cancer are shown in **Table 1** by the type of familial constellation. Family history of any hepatobiliary cancer showed a relative risk of 1.75, marginally higher for women (1.82) than for men (1.70). The sex difference was somewhat higher when both the index case (offspring) and the proband (parent) were of the same sex, females (2.02) and males (1.64). The risks between siblings (2.08) were somewhat higher than between offspring and parents (1.59), or when two instead of one family members were probands.

Analysis by specific hepatobiliary cancer is shown in **Table 2** when a family member was diagnosed with any hepatobiliary cancer. For HCC, the overall SIR was 2.16, slightly higher than for GBC (2.02). For intrahepatic bile duct cancer, the risk was significant only for men (SIR 1.70). No familial risk was observed for extrahepatic bile duct and ampullary cancers. When ICD-10 data (from 1997 onwards, in which HCC and intrahepatic bile duct cancers were separated) were used for case identification, familial risks were somewhat higher and the SIR of 1.87 for intrahepatic bile ducts was significant.

In **Table 3**, analysis focused on concordant and discordant hepatobiliary cancer types. Concordant familial risks were almost equally high for HCC (2.60) and GBC (2.76). Significant association of extrahepatic bile duct cancer was only noted with GBC (2.85). Results for (concordant) intrahepatic bile duct cancer could be analyzed based on the ICD-10 with follow up from 1997 onwards, and the results are shown in the bottom part of the table. A concordant association for intrahepatic bile duct cancer was observed (3.81); no discordant associations were observed. Concordant association of HCC (4.06) was higher than that in the upper part of the table because the observation time was only 22 years and familial pairs were likely to be relatively young siblings. No discordant associations were observed. Concordant association for GBC was of borderline significance (2.14) but the discordant association with cancer of the extrahepatic bile ducts (3.71) was significant. Related sex-specific data are shown in **Appendix A**. The new observation was the female association of HCC with GBC (1.73).

In **Table 4** familial risks were analyzed when family members were diagnosed with any cancer. HCC was associated with lung (1.36) and cervical (1.27) cancers. GBC was associated with no discordant cancer but extrahepatic bile duct and ampullary cancers shared associations with pancreatic and colon cancers. Extrahepatic bile duct cancer was additionally associated with small intestinal cancer and melanoma, while ampullary cancer showed a high association (2.20) with nervous system cancer. Among these 13 patients, 6 had an affected sibling and 7 an affected parent; their diagnoses were 6 astrocytomas (5 grade III-IV, 1 grade I-II), 4 meningiomas and the rest individual diverse histologies. Related sex-specific data are shown in **Appendix A**. The new observations were the male association of GBC with lung cancer (1.48) and ampullary cancer with male breast cancer (1.45).

Analysis of combined hepatobiliary cancers between spouses showed a risk of 1.29 for wives when their husbands were diagnosed with hepatobiliary cancer (**Table 5**). This was largely explained by HCC risk of 1.33. Risk in husbands was not significant for any hepatobiliary cancer. We analyzed the risk of hepatobiliary cancer when the spouse was diagnosed with any cancer and in **Table 5** results are shown for sites with significant associations. Risk in wives was increased when their husbands were diagnosed with stomach (1.49) or pancreatic (1.26) cancer; risk in husbands was increased (1.19) when their wives were diagnosed with cervical cancers. Association with upper aerodigestive tract cancer was significant when either spouse was affected.

Rate ratios of familial and nonfamilial HCC and GBC were defined for age-specific incidence rates (**Figure 2**). For both cancers, the familial incidence peaked at age 75–79 years, the rate ratios being 2.16 and 2.25, respectively. No prominent early-onset components were apparent.

The risk factors (comorbidities) for hepatobiliary cancer are listed in **Appendix A**. Chronic obstructive pulmonary disease was used as a surrogate of smoking, as no smoking data were available. Notably, 2.9 million individuals had been diagnosed with some risk factor, the most common of which were chronic obstructive pulmonary disease (0.96 million) and diabetes (0.76 million). Familial risks were calculated for hepatobiliary cancer in offspring whose first-degree family members were diagnosed with any of the 12 comorbidities (**Table 6**). Association with HBV showed the highest SIR of 4.05, followed by alcohol-related liver disease (1.60), chronic obstructive pulmonary disease (1.14) and diabetes (1.13). Alcohol-related liver disease and diabetes were associated with cancer in both sexes, but male risks exceed female risks. Altogether, 4046 hepatobiliary cancers were observed, and more than half were in association with diabetes.

Similar analysis was conducted by type of hepatobiliary cancer (**Table 7**). HCC dominated in this analysis, and all significant risk estimates from **Table 6** were more significant in **Table 7**. A novel significant association was noted for primary biliary cirrhosis (2.58). The only association with non-HCC sites was for ampullary cancer (1.29), associated with diabetes.

## 4. Discussion

Family studies in hepatobiliary cancers in a low-risk-factor setting, such as Sweden, are not common, particularly with verified case and family histories; the present study is the largest published so far. Familial proportions of these cancers were low: 3.6% HCC, 4.0% GBC, 2.5% extrahepatic bile duct cancer and 2.0% ampullary cancer. In addition to HCC, we were also able to assess familial risks for other hepatobiliary cancers and their association with any other cancers from the same dataset, which is unique. Analysis of cancers among spouses was informative of shared environmental factors and showed an increased risk of hepatobiliary cancer when the other spouse was diagnosed with HCC, stomach, pancreatic, cervical or upper aerodigestive tract cancers.

Familial relative risk for concordant HCC was 2.60, somewhat higher for men (2.73) than for women (2.32). These are at the level of a previous meta-analysis [18]. HCC did not show associations with other, much rarer hepatobiliary sites, with the exception of female risk with GBC. HCC was associated with lung and cervical cancer, which together with spouse risks points to a lifestyle featuring tobacco, alcohol and broken marriages [30,31]. The incidence between familial and nonfamilial rates were rather uniform over the age range up to 80 years. This suggests that the underlying risk factors pose risk over a large part of the lifetime. This may be different from single-gene effects, which may show an early age-defined peak (**Figure 2**).

We showed concordant familial risks for GBC (2.76), based on ICD-10 for intrahepatic bile duct cancers (3.81) (**Table 3**). Concordant familial association for extrahepatic bile duct cancers was not significant but the risk was significant in the ICD-10 data for women (2.23) when probands were diagnosed with any hepatobiliary cancer; male risk in this analysis was only 1.19 (95% CI 0.47–2.47) (**Table 2**). In the same analysis, male intrahepatic bile duct cancer risk was significant (2.50) but female risk was much lower (1.24; 95% CI 0.53–2.45). The only discordant association was between GBC and extrahepatic bile duct cancer, and the SIR (2.85) was at the level of concordant risk for GBC (2.76) (**Table 3**). We defined incidence rates between familial and nonfamilial GBC (**Figure 2**). Although the familial incidence curve was fairly similar to that of HCC, there was a peak at age 55–60 years which would coincide with GBC incidence in Lynch syndrome [26,27]. Previous literature on familial risks at these sites is scanty, and in addition to our results of GBC, most literature stems from studies in Asia, from different risk factor settings [4,21,22,23]. We recently published a study on second primary cancers, after hepatobiliary cancers and hepatobiliary cancers, as second primaries after any cancers [32]. HCC and GBC associated with each other, as did GBC and bile duct cancer, while HCC and bile duct cancer showed no associations. Thus, these patterns agree with the present familial risks.

Analysis of familial associations of GBC and bile duct cancers, with cancers at extrahepatic sites, showed no significant associations for GBC. Extrahepatic bile duct cancers were associated with small intestinal, pancreatic and colon cancers and melanoma; ampullary cancers shared associations with pancreatic and colon cancers, and were additionally associated with nervous system cancers (with expected frequencies of gliomas and meningiomas, suggesting that the association was fortuitous) [33]. Ampullary cancers were associated also with male breast cancers. Associations between anatomically adjacent sites may be due to diagnostic bias but may also be due to physiological links. Bile flows from the liver through the bile ducts and gall bladder into the small intestine and then into the colon. The pancreatic duct joins the bile duct before the ampulla of Vater. Bile stones may cause backflow of bile into the pancreas. Thus, carcinogens in bile would have access to all of these organs. Biliary tract cancers may harbor mismatch repair gene mutations which may explain the associations with colon and pancreatic cancers [26,27]. Recently, these mutations have also been found in ampullary cancer [34]. The association of ampullary cancers with male breast cancer may be related to BRCA1/2 mutations which are found in this cancer [35].

Familial associations between hepatobiliary cancers and their risk factors were dominated by HCC; the only association with non-HCC sites was for ampullary cancer associating with diabetes. Among HCC risk factors, familial association with HBV was high but the case numbers were few, in contrast to alcohol with modest risk and high number of affected relatives. HCC risk was barely over 1.1 when family members presented with common risk factors, diabetes or chronic obstructive pulmonary disease. These data suggest that shared risk factors in family appear not to explain much of familial risk of hepatobiliary cancers, and rather that the risk factors are individual.

The inherent limitation of studying familial risks in rare cancers is the low case numbers and low statistical power. This is evident in global literature in familial hepatobiliary cancers, with a volume of papers on HCC particularly from studies carried out in Asian countries where HCC is common, but there is hardly any extant literature on GBC and bile duct cancers. The power of this study was that we could address even these rare cancers in an unbiased manner. In the analysis of risk factors, identification of patients was based on diagnostic codes which were changed, or new codes were introduced over time implying inaccuracies and missing cases. Particularly, for smoking and alcohol, moderate users were missed.

## 5. Conclusions

Within the available sample size, we showed that concordant familial risks of HCC, GBC and intrahepatic and extrahepatic bile duct cancers are relatively high, between 2.5 and 3.0, compared with other cancers [36]. While familial risks in HCC and GBC did not show large sex differences, bile duct cancers may show large sex differences, but larger studies are needed to confirm this. There appeared to be limited sharing of familial risks between these sites, confined to HCC and GBC, and to GBC and extrahepatic bile duct cancers. Environmental factors may be more important causes of familial clustering of these cancers than for many other familial cancers, for which inherited genetic factors may play an important role [36]. Yet, among risk factors in family members, alcohol-related disease appeared to be an important contributor to familial risk of HCC only. While there are several drugs approved for treatment of hepatobiliary cancers, curative treatment of metastatic and advanced disease remains difficult. Therefore, considering the overall poor prognosis of these cancers, prevention is of the utmost importance, including moderation of alcohol consumption, prevention of HBV infection with vaccine, treatment of HBV and HCV infections and avoidance of overweight and other risk factors of type 2 diabetes.

## Figures and Tables

**Figure 1 cancers-14-01938-f001:**
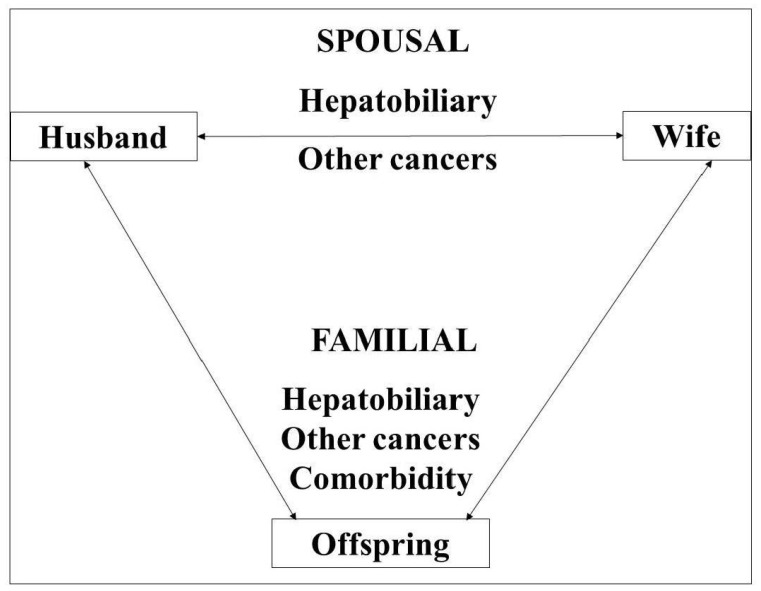
The scheme for the current analysis highlighting the study subjects (parent–offspring and spouses), the types of analysis (familial and spousal) and the diseases (hepatobiliary and other cancers, and comorbidities of hepatobiliary cancer) considered.

**Figure 2 cancers-14-01938-f002:**
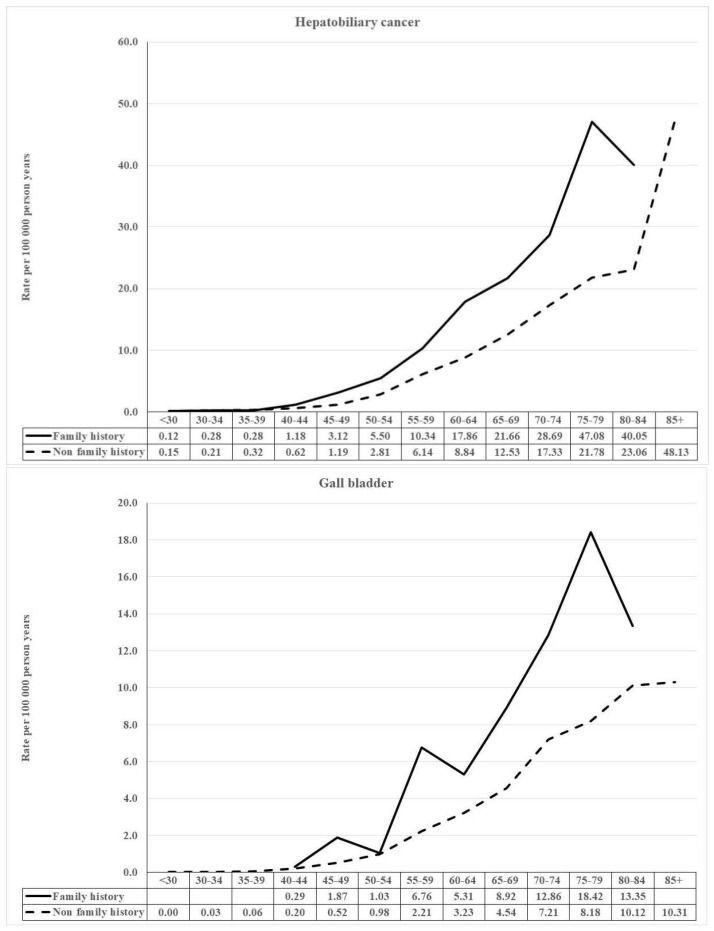
Age-specific incidence for familial and nonfamilial hepatocellular carcinoma and gallbladder cancer with incidence rate ratios.

**Table 1 cancers-14-01938-t001:** Familial risks for concordant hepatobiliary cancer.

Family Relation	Men				Women				All			
	O	SIR	95% CI		O	SIR	95% CI		O	SIR	95% CI	
Family history (any first-degree relatives)	168	**1.70**	**1.45**	**1.98**	136	**1.82**	**1.53**	**2.15**	304	**1.75**	**1.56**	**1.96**
Parents	110	**1.48**	**1.22**	**1.78**	95	**1.73**	**1.40**	**2.12**	205	**1.59**	**1.38**	**1.82**
Father	52	**1.64**	**1.23**	**2.15**	33	1.37	0.95	1.93	85	**1.53**	**1.22**	**1.89**
Mother	58	**1.38**	**1.05**	**1.79**	64	**2.02**	**1.55**	**2.58**	122	**1.66**	**1.37**	**1.98**
Siblings	59	**2.09**	**1.59**	**2.69**	43	**2.07**	**1.50**	**2.79**	102	**2.08**	**1.69**	**2.52**
One family member	167	**1.67**	**1.43**	**1.95**	132	**1.78**	**1.49**	**2.11**	299	**1.72**	**1.53**	**1.92**
Two family members	1	1.01	0.00	5.81	4	**5.85**	**1.52**	**15.13**	5	2.99	0.94	7.04

O—observed; SIR—standardized incidence ratio; CI—confidence intervals. Bold types: 95% CI does not include 1.00.

**Table 2 cancers-14-01938-t002:** Familial risks of hepatobiliary cancer when a first-degree relative had any type of hepatobiliary cancer.

Cancer Type	Men				Women				All			
	O	SIR	95% CI		O	SIR	95% CI		O	SIR	95% CI	
**Diagnosis of cases based on ICD-7, 1958–2018**												
Hepatobiliary cancer	180	**1.80**	**1.55**	**2.09**	139	**1.85**	**1.56**	**2.18**	319	**1.82**	**1.63**	**2.04**
Subtypes of hepatobiliary cancer in the study population												
HCC	120	**2.01**	**1.67**	**2.41**	49	**2.64**	**1.96**	**3.50**	169	**2.16**	**1.85**	**2.51**
Gallbladder cancer	17	**1.90**	**1.10**	**3.04**	55	**2.06**	**1.55**	**2.68**	72	**2.02**	**1.58**	**2.54**
Intrahepatic bile duct cancer	21	**1.70**	**1.05**	**2.60**	14	1.04	0.56	1.74	35	1.35	0.94	1.88
Extrahepatic bile duct cancer	13	1.08	0.57	1.85	17	1.57	0.91	2.52	30	1.31	0.88	1.87
Ampulla of Vater cancer	9	1.33	0.60	2.53	4	0.71	0.19	1.85	13	1.05	0.56	1.80
**Diagnosis of cases based on ICD-10, 1997–2018**												
Hepatobiliary cancer	101	**2.02**	**1.65**	**2.46**	69	**1.97**	**1.53**	**2.50**	170	**2.00**	**1.71**	**2.32**
Subtypes of hepatobiliary cancer in the study population												
HCC	68	**2.28**	**1.77**	**2.89**	26	**2.83**	**1.84**	**4.15**	94	**2.41**	**1.95**	**2.95**
Gallbladder	4	0.83	0.22	2.14	21	**1.76**	**1.09**	**2.69**	25	1.49	0.96	2.20
Intrahepatic bile duct	16	**2.50**	**1.42**	**4.07**	8	1.24	0.53	2.45	24	**1.87**	**1.19**	**2.78**
Extrahepatic bile duct	7	1.19	0.47	2.47	11	**2.23**	**1.11**	**4.01**	18	1.67	0.99	2.64
Ampulla of Vater	6	1.96	0.70	4.29	3	1.22	0.23	3.63	9	1.63	0.74	3.11

O—observed; SIR—standardized incidence ratio; CI—confidence intervals. Bold types: 95% CI does not include 1.00.

**Table 3 cancers-14-01938-t003:** Familial risks of concordant and discordant hepatobiliary cancer types.

**Cancer in Family**	**Diagnoses Based on ICD-7, 1958–2018**																			
	**HCC**				**Gallbladder**				**Extrahepatic Bile Ducts**					**Ampulla of Vater**						
	**O**	**SIR**	**95% CI**		**O**	**SIR**	**95% CI**		**O**	**SIR**		**95% CI**		**O**	**SIR**	**95% CI**				
HCC	122	**2.60**	**2.16**	**3.10**	24	1.43	0.92	2.13	10	0.92		0.44	1.71	6	1.38	0.50			3.03	
Gallbladder	50	1.32	0.98	1.74	39	**2.76**	**1.96**	**3.77**	12	1.30		0.67	2.28	2	0.55	0.05			2.02	
Extrahepatic bile ducts	10	0.81	0.39	1.50	13	**2.85**	**1.51**	**4.89**	6	2.04		0.73	4.47	2	1.70	0.16			6.25	
Ampulla of Vater	7	1.33	0.53	2.75	0				1	0.81		0.00	4.67	0						
All	189	**1.85**	**1.59**	**2.13**	76	**2.04**	**1.60**	**2.55**	29	1.20		0.80	1.72	10	1.04	0.49			1.91	
**Cancer in Family**	**Diagnoses based on ICD-10, 1997–2018**																			
	**HCC**				**Intrahepatic bile ducts**				**Gallbladder**				**Extrahepatic bile ducts**				**Ampulla of Vater**			
	**O**	**SIR**	**95% CI**		**O**	**SIR**	**95% CI**		**O**	**SIR**	**95% CI**		**O**	**SIR**	**95% CI**		**O**	**SIR**	**95% CI**	
HCC	70	**4.06**	**3.16**	**5.13**	8	1.42	0.61	2.81	7	0.95	0.38	1.98	4	0.85	0.22	2.20	6	2.51	0.90	5.50
Intrahepatic bile ducts	5	1.08	0.34	2.54	6	**3.81**	**1.37**	**8.35**	4	1.98	0.52	5.12	1	0.76	0.00	4.37	3	4.50	0.85	13.32
Gallbladder	11	1.13	0.56	2.03	6	1.89	0.68	4.13	9	2.14	0.97	4.08	10	**3.71**	**1.77**	**6.85**	0			
Extrahepatic bile ducts	4	0.86	0.22	2.21	3	1.91	0.36	5.66	5	2.44	0.77	5.74	2	1.51	0.14	5.56	0			
Ampulla of Vater	4	1.47	0.38	3.81	1	1.11	0.00	6.37	0				1	1.31	0.00	7.48	0			
All	94	**2.41**	**1.95**	**2.95**	24	**1.87**	**1.19**	**2.78**	25	1.49	0.96	2.20	18	1.67	0.99	2.64	9	1.63	0.74	3.11

O—observed; SIR—standardized incidence ratio; CI—confidence intervals. Bold types: 95% CI does not include 1.00.

**Table 4 cancers-14-01938-t004:** Familial risks of hepatobiliary cancer when family members had any cancer.

Cancer in Family	HCC				Gallbladder				Extrahepatic Bile Ducts				Ampulla of Vater			
	O	SIR	95% CI		O	SIR	95% CI		O	SIR	95% CI		O	SIR	95% CI	
Upper aerodigestive tract	117	1.15	0.95	1.38	43	1.19	0.86	1.61	25	1.06	0.69	1.57	11	1.17	0.58	2.09
Salivary gland	10	0.95	0.45	1.75	2	0.55	0.05	2.03	2	0.83	0.08	3.05	2	2.02	0.19	7.44
Esophagus	44	1.17	0.85	1.57	21	1.58	0.98	2.42	3	0.35	0.07	1.03	3	0.86	0.16	2.53
Stomach	152	0.95	0.80	1.11	73	1.22	0.96	1.54	41	1.05	0.75	1.42	21	1.37	0.85	2.10
Small intestine	19	0.95	0.57	1.48	6	0.86	0.31	1.89	10	**2.18**	**1.04**	**4.02**	4	2.16	0.56	5.59
Pancreas	120	1.14	0.95	1.37	46	1.21	0.89	1.61	41	**1.65**	**1.18**	**2.24**	18	**1.80**	**1.07**	**2.85**
Colon	300	0.95	0.85	1.07	103	0.92	0.75	1.11	94	**1.28**	**1.03**	**1.57**	48	**1.60**	**1.18**	**2.13**
Rectum	156	0.96	0.81	1.12	70	1.20	0.94	1.52	33	0.87	0.60	1.22	16	1.04	0.59	1.69
Anus	9	1.02	0.46	1.95	3	1.01	0.19	3.00	2	1.03	0.10	3.80	0			
Lung	384	**1.36**	**1.23**	**1.51**	102	1.04	0.85	1.26	67	1.05	0.82	1.34	29	1.12	0.75	1.62
Hepatobiliary	189	**1.85**	**1.59**	**2.13**	76	**2.04**	**1.60**	**2.55**	29	1.20	0.80	1.72	10	1.04	0.49	1.91
Breast	395	0.90	0.82	1.00	158	1.04	0.88	1.21	97	0.98	0.79	1.19	47	1.14	0.84	1.52
Cervix	71	**1.27**	**1.00**	**1.61**	18	0.94	0.55	1.48	7	0.56	0.22	1.17	7	1.40	0.56	2.90
Endometrium	108	1.07	0.87	1.29	35	0.98	0.68	1.36	17	0.73	0.42	1.17	7	0.74	0.29	1.53
Ovary	60	0.87	0.67	1.12	21	0.87	0.54	1.33	17	1.07	0.62	1.72	3	0.46	0.09	1.37
Other female genitals	10	0.67	0.32	1.25	8	1.49	0.64	2.96	4	1.15	0.30	2.97	0			
Prostate	419	0.91	0.82	1.00	147	0.91	0.76	1.06	76	0.71	0.56	0.89	27	0.62	0.41	0.90
Testis	11	1.41	0.70	2.54	1	0.41	0.00	2.36	2	1.26	0.12	4.65	1	1.54	0.00	8.85
Other male genitals	6	1.14	0.41	2.50	0	0.00	0.53	2.17	1	0.84	0.00	4.80	0			
Kidney	75	1.00	0.79	1.25	26	0.97	0.63	1.42	20	1.15	0.70	1.78	9	1.28	0.58	2.44
Bladder	124	1.01	0.84	1.20	48	1.12	0.82	1.48	24	0.85	0.55	1.27	9	0.78	0.35	1.48
Melanoma	84	0.99	0.79	1.22	33	1.17	0.80	1.64	28	**1.52**	**1.01**	**2.19**	8	1.02	0.44	2.03
Skin	96	0.84	0.68	1.03	49	1.23	0.91	1.63	29	1.12	0.75	1.61	11	1.00	0.50	1.80
Eye	7	1.07	0.42	2.21	4	1.72	0.45	4.44	2	1.32	0.12	4.84	0			
Nervous system	66	1.02	0.79	1.30	24	1.10	0.70	1.64	18	1.25	0.74	1.99	13	**2.20**	**1.17**	**3.78**
Thyroid	20	1.18	0.72	1.82	4	0.68	0.18	1.77	1	0.26	0.00	1.51	2	1.31	0.12	4.82
Endocrine	41	1.11	0.79	1.50	15	1.21	0.67	1.99	5	0.61	0.19	1.44	4	1.20	0.31	3.11
Bone	0	0.00	0.29	1.19	2	1.85	0.17	6.81	1	1.41	0.00	8.09	0			
Connective tissue	17	1.24	0.72	2.00	2	0.43	0.04	1.58	2	0.64	0.06	2.37	0			
Hodgkins disease	6	0.62	0.22	1.36	5	1.50	0.47	3.52	3	1.37	0.26	4.04	3	3.45	0.65	10.21
Non-Hodgkins lymphoma	53	0.80	0.60	1.05	31	1.35	0.92	1.92	12	0.80	0.41	1.40	5	0.81	0.26	1.91
Myeloma	39	1.26	0.90	1.73	14	1.26	0.69	2.12	4	0.55	0.14	1.43	1	0.34	0.00	1.96
Leukemia	66	0.97	0.75	1.24	22	0.94	0.59	1.43	10	0.65	0.31	1.20	4	0.64	0.17	1.64
Primary unknown	67	1.06	0.82	1.35	22	0.98	0.61	1.49	19	1.30	0.78	2.04	5	0.85	0.27	2.00
Others	5	1.01	0.32	2.37	1	0.60	0.00	3.42	1	0.88	0.00	5.06	1	2.20	0.00	12.60
All	3377	**1.04**	**1.00**	**1.07**	1244	**1.08**	**1.02**	**1.15**	753	1.00	0.93	1.08	330	1.08	0.97	1.20

O—observed; SIR—standardized incidence ratio; CI—confidence intervals. Bold types: 95% CI does not include 1.00.

**Table 5 cancers-14-01938-t005:** Risks of hepatobiliary cancer among spouses.

**Hepatobiliary Cancer in Spouse**	**Wives**				**Husbands**				**All**			
	**O**	**SIR**	**95% CI**		**O**	**SIR**	**95% CI**		**O**	**SIR**	**95% CI**	
Hepatobiliary cancer	97	**1.29**	**1.04**	**1.57**	97	1.12	0.91	1.37	194	**1.20**	**1.04**	**1.38**
HCC	64	**1.33**	**1.02**	**1.70**	37	1.33	0.94	1.84	101	**1.33**	**1.08**	**1.62**
Gallbladder	17	1.30	0.75	2.08	46	1.04	0.76	1.39	63	1.10	0.85	1.41
Extrahepatic bile ducts	10	1.09	0.52	2.01	9	0.86	0.39	1.64	19	0.97	0.58	1.51
Ampulla of vater	6	1.22	0.44	2.67	5	1.20	0.38	2.83	11	1.21	0.60	2.18
**Other cancers in spouse**	**Wives**				**Husbands**				**All**			
	**O**	**SIR**	**95% CI**		**O**	**SIR**	**95% CI**		**O**	**SIR**	**95% CI**	
Upper aerodistive trac	130	1.13	0.95	1.34	56	1.25	0.94	1.62	186	**1.16**	**1.00**	**1.34**
Stomach	208	**1.49**	**1.30**	**1.71**	83	0.92	0.74	1.15	291	**1.27**	**1.13**	**1.43**
Pancreas	123	**1.26**	**1.04**	**1.50**	90	0.92	0.74	1.13	213	1.09	0.95	1.25
Cervix					138	**1.19**	**1.00**	**1.40**	138	**1.19**	**1.00**	**1.40**

O—observed; SIR—standardized incidence ratio; CI—confidence intervals. Bold types: 95% CI does not include 1.00.

**Table 6 cancers-14-01938-t006:** Familial risks of hepatobiliary cancer when family members were diagnosed with comorbidities, 1964–2018.

Diagnosis of Comorbidities in Family	Men				Women				All			
	O	SIR	95% CI		O	SIR	95% CI		O	SIR	95% CI	
Alcohol-related liver disease	345	**1.79**	**1.61**	**1.99**	196	**1.34**	**1.16**	**1.54**	541	**1.60**	**1.47**	**1.74**
Chronic obstructive pulmonary disease	676	**1.17**	**1.09**	**1.27**	478	1.08	0.99	1.19	1154	**1.14**	**1.07**	**1.20**
Gallstone disease or operation	3	0.41	0.08	1.23	6	1.17	0.42	2.55	9	0.73	0.33	1.39
Hepatitis B virus	4	**3.90**	**1.01**	**10.09**	3	4.28	0.81	12.66	7	**4.05**	**1.61**	**8.40**
Hepatitis C virus	1	0.28	0.00	1.60	2	0.76	0.07	2.78	3	0.48	0.09	1.42
Hepatitis of any kind	12	1.49	0.77	2.61	5	0.84	0.26	1.97	17	1.21	0.71	1.95
Infection of bile ducts	45	1.08	0.79	1.45	38	1.16	0.82	1.60	83	1.12	0.89	1.39
Autoimmune hepatitis	27	1.21	0.80	1.77	13	0.77	0.41	1.32	40	1.02	0.73	1.39
Non-alcohol-related liver disease	2	1.17	0.11	4.30	1	0.87	0.00	5.00	3	1.05	0.20	3.11
Obesity	50	1.32	0.98	1.74	30	1.08	0.73	1.54	80	1.22	0.96	1.51
Primary biliary cirrhosis	5	1.67	0.53	3.92	4	1.91	0.50	4.94	9	1.77	0.80	3.37
Diabetes	1238	**1.17**	**1.11**	**1.24**	862	**1.07**	**1.00**	**1.15**	2100	**1.13**	**1.08**	**1.18**
All of above	2408	**1.23**	**1.19**	**1.28**	1638	**1.10**	**1.05**	**1.16**	4046	**1.18**	**1.14**	**1.21**

O—observed; SIR—standardized incidence ratio; CI—confidence intervals. Bold types: 95% CI does not include 1.00.

**Table 7 cancers-14-01938-t007:** Familial risks of specific hepatobiliary cancer when family members were diagnosed with comorbidities, 1964–2018.

Diagnosis of Comorbidities in Family	HCC				Gallbladder				Extrahepatic Bile Ducts				Ampulla of Vater			
	O	SIR	95% CI		O	SIR	95% CI		O	SIR	95% CI		O	SIR	95% CI	
Alcohol-related liver disease	414	**2.00**	**1.81**	**2.20**	63	0.95	0.73	1.22	47	1.00	0.74	1.34	17	0.91	0.53	1.46
Chronic obstructive pulmonary disease	734	**1.23**	**1.14**	**1.32**	217	1.00	0.87	1.14	149	1.03	0.87	1.21	54	0.91	0.69	1.19
Gallstone disease or operation	5	0.68	0.22	1.60	3	1.17	0.22	3.47	1	0.56	0.00	3.23	0			
Hepatitis B viru	7	**5.20**	**2.06**	**10.77**	0				0				0			
Hepatitis C virus	3	0.69	0.13	2.03	0				0				0			
Hepatitis of any kind	13	1.58	0.84	2.71	3	1.01	0.19	2.99	0				1	1.22	0.00	7.00
Infection of bile ducts	51	1.22	0.91	1.60	17	1.00	0.58	1.60	6	0.56	0.20	1.22	9	2.01	0.91	3.84
Autoimmune hepatitis	28	1.23	0.81	1.78	6				4	0.71	0.19	1.85	2			
Non-alcohol-related liver disease	2	1.07	0.10	3.95	1	2.21	0.00	12.65	0				0			
Obesity	55	1.28	0.97	1.67	17	1.51	0.88	2.42	4	0.47	0.12	1.22	4	1.26	0.33	3.25
Primary biliary cirrhosis	8	**2.58**	**1.10**	**5.12**	0				1	1.36	0.00	7.82	0			
Diabetes	1271	**1.18**	**1.12**	**1.25**	418	1.02	0.93	1.13	281	1.05	0.93	1.18	130	**1.20**	**1.00**	**1.43**
All of above	2591	**1.29**	**1.24**	**1.34**	745	1.01	0.94	1.09	493	1.01	0.92	1.10	217	1.09	0.95	1.25

HCC—hepatocellular carcinoma; O—observed; SIR—standardized incidence ratio; CI—confidence intervals. Bold types: 95% CI does not include 1.00.

## Data Availability

Data used were issued by the National Board of Health and Welfare, Stockholm, to Kristina and Jan Sundquist for exclusive institutional use. Any data requests should be directed to the National Board of Health and Welfare, Stockholm.

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
