# Peer review of "Familial Risks for Liver, Gallbladder and Bile Duct Cancers and for Their Risk Factors in Sweden, a Low-Incidence Country"

_cancers, 2022, doi:10.3390/cancers14081938_

Round 1
Reviewer 1 Report
In this manuscript, the authors evaluated familial risks for concordant (same) and discordant (different) hepatobiliary cancers, including their associations with any other cancers utilizing resources from the Swedish Cancer Registry data. They used standardized incidence ratios (SIRs) to estimate adjusted familial risks and found an SIR of 2.60 and 2.76 for concordant hepatocellular carcinoma (HCC) and gallbladder cancer, respectively. They also found association of HCC with other cancer types and that family members diagnosed with alcohol related disease were associated with HCC.
The manuscript is well-written, the methods section logically follows, in parallel, with the results section. The discussion provides a clear narrative of the potential importance of this study and provides a thorough description of the limitations. I only have several minor comments.
- The introduction section, although informative, is the way too long. The authors may consider shortening it.
- In the methods section, since so many types of liver cancer and their associations with other cancer types were investigated, a figure or diagram showing all analyses included in this study may be helpful for readers to follow.
- Lines 273-279. There are misused line breaks.
- In conclusions, I disagree the use of references.
- References: Inconsistent reference forms. Some reference use lower cases and full names (e.g., ref 7. International journal of cancer) for journal names, while others use capital letters and abbreviations (e.g., ref 8. Clin Gastroenterol Hepatol).
Author Response
The manuscript is well-written, the methods section logically follows, in parallel, with the results section. The discussion provides a clear narrative of the potential importance of this study and provides a thorough description of the limitations. I only have several minor comments.
>>> Thank you, language was edited.
The introduction section, although informative, is the way too long. The authors may consider shortening it.
>>> Most of the first paragraph was deleted and merged with the second paragraph.
In the methods section, since so many types of liver cancer and their associations with other cancer types were investigated, a figure or diagram showing all analyses included in this study may be helpful for readers to follow.
>>> A new figure 1 with a legend was introduced, with appropriate text in Methods.
Lines 273-279. There are misused line breaks.
>>> This was not in the original manuscript.
In conclusions, I disagree the use of references.
References: Inconsistent reference forms. Some reference use lower cases and full names (e.g., ref 7. International journal of cancer) for journal names, while others use capital letters and abbreviations (e.g., ref 8. Clin Gastroenterol Hepatol
>>> This is an Endnote problem even though correct abbreviations are in the database. It has to be fixed at proofs stage.
Reviewer 2 Report
This manuscript is an original article that investigated familial risks for hepatobiliary cancers using the Swedish Cancer Registry data. The authors analyzed large number of data and presented them in detail.
This study was conducted well, and the methods are appropriate. The data are presented clearly. I think this is a well-written paper.
The results will be of interest to clinicians and researchers in the field.
Author Response
This manuscript is an original article that investigated familial risks for hepatobiliary cancers using the Swedish Cancer Registry data. The authors analyzed large number of data and presented them in detail.
This study was conducted well, and the methods are appropriate. The data are presented clearly. I think this is a well-written paper.
The results will be of interest to clinicians and researchers in the field.
We thank the reviewer!